# Neurobehavioral evidence of interoceptive sensitivity in early infancy

Lara Maister[1,2,3]*, Teresa Tang[1], Manos Tsakiris[1,2]

[1]Lab of Action and Body, Department of Psychology, Royal Holloway University of London, Surrey, United Kingdom; [2]The Warburg Institute, School of Advanced Study, University of London, London, United Kingdom; [3]Department of Psychological Sciences, Birkbeck University of London, London, United Kingdom

**Abstract** Interoception, the sensitivity to visceral sensations, plays an important role in homeostasis and guiding motivated behaviour. It is also considered to be fundamental to self-awareness. Despite its importance, the developmental origins of interoceptive sensitivity remain unexplored. We here provide the first evidence for implicit, flexible interoceptive sensitivity in 5 month old infants using a novel behavioural measure, coupled with an established cortical index of interoceptive processing. These findings have important implications for the understanding of the early developmental stages of self-awareness, self-regulation and socio-emotional abilities.

## Main text

Our entire experience of the world is perceived against a pervasive backdrop of internal physiological sensations from our bodies. Our sensitivity to these visceral sensations, known as 'interoception', is thought to be the fundamental basis for subjective feeling states (*Craig, 2009*; *Damasio, 2010*), and important for maintaining homeostasis (*Gu and FitzGerald, 2014*). Interoceptive sensitivity is a stable trait which varies between individuals, and influences a range of psychological processes from emotion processing to decision-making and various psychological disorders (*Critchley and Harrison, 2013*). Inevitably, interoception has recently witnessed an exponential rise in interest across the neurosciences, psychology and psychiatry (*Khalsa and Lapidus, 2016*).

Behaviourally, individual differences in interoception are commonly measured by assessing participants' ability to count their own heartbeats (*Schandry, 1981*) or discriminate cardio-auditory synchrony from asynchrony (*Brener et al., 1993*). Interoceptive processing can also be observed in the electroencephalogram (EEG); the Heartbeat Evoked Potential (HEP) is an electrophysiological index of cortical cardiac processing, and is found between 250–600 ms after the cardiac R wave over frontocentral (*Pollatos and Schandry, 2004*) and parietal regions (*Dirlich et al., 1998*; *Couto et al., 2015*). HEP amplitude is positively correlated with interoception at rest (*Pollatos and Schandry, 2004*), and is also modulated by *state* changes in interoception such as during emotional processing (*Fukushima et al., 2011*).

Beyond its fundamental role in adulthood, interoception has been given prominence in developmental theories, which highlight its importance in the infant's experience of reward, motivation and arousal (*Fogel, 2011*; *Mundy and Jarrold, 2010*; *Fotopoulou and Tsakiris, 2017*). Interoceptive sensitivity may allow infants to perform crucial self-regulatory behaviours that facilitate homeostasis. By linking perceived internal sensations with events or objects in the environment, infants may be able to develop allostatic seeking or avoiding behaviours depending on the nature of the internal sensation. For example, accurately perceiving sensations of satiety and linking these sensations with feeding behaviour may allow infants to self-regulate their milk intake (*Harshaw, 2008*). Furthermore, the detection of unpleasant arousal states may initiate attentional disengagement or avoidance of

*For correspondence: lara.maister@rhul.ac.uk

**Competing interests:** The authors declare that no competing interests exist.

**eLife digest** From the beginning till the end of a person's life, parts of the body continuously send signals to the brain. Most of this happens without the person even being aware of it, yet people can become aware of the signals under certain circumstances. For example, we can feel our racing heart rate or the "butterflies in our stomach" when we are anxious or excited. This ability to consciously sense signals from the body is called interoception, and some people are more aware of these signals than others. These differences between people can influence a wide range of psychological processes, including how strongly they feel emotions, how they make decisions, and their mental health.

Despite the crucial role that interoception plays in thought processes in adults, scientists know practically nothing about how it first develops. Progress in this field has been hindered largely because there was no way to measure sensitivity to interoceptive signals in infants.

Now, Maister et al. have developed a new task called iBEATS that can measure how sensitive an infant is to their own heartbeat. During the task, five-month old infants were shown an animated character that either moved in synchrony with their own heartbeat or out of synchrony with their heartbeat. The infants spent longer looking at the character that was moving out of synchrony than the one moving in synchrony, suggesting that even at this early age, infants can sense their own interoceptive signals.

As with adults, some of the infants were more sensitive to their heartbeats than others, and Maister et al. could see these differences played out in the infant's brain activity via electrodes placed on the infant's head. Infants who had shown a strong preference in the iBEATS task also showed a larger brain signal known as the Heart-Evoked Potential (or HEP). Furthermore, this brain signal got larger when infants viewed a video clip of an angry or fearful face. This suggests that the infants' brains were monitoring their hearts more closely when they were confronted with negative emotions.

This study provides a validated measure of interoception for very young participants. Using this task, researchers can now investigate which factors affect how awareness to interoceptive signals develops, including social interactions and the infant's temperament. Maister et al. also plan to carry out longer-term experiments to learn exactly how interoception may influence the development of emotional abilities, and also what role it might play in disorders such as anxiety and depression. The findings of these future experiments may eventually guide interventions to treat these conditions.

an aversive stimulus. Therefore, individual differences in interoception may represent an important determinant in socio-emotional and physical development. However, despite much speculation regarding the developmental origins of interoception, developmental research in this area has been restricted to older childhood (*Koch and Pollatos, 2014*) due to methodological limitations. Here, we develop a novel behavioural measure of interoception in infancy, and present the first ever demonstration of interoceptive sensitivity at 5 months of age, coupled with electrophysiological markers, to show that (a) infants display an implicit sensitivity to interoceptive signals, and (b), that this sensitivity is responsive to socio-emotional processing demands, as in adults.

We developed a novel implicit behavioural measure, the *Infant Heartbeat Task (iBEAT)*, which employs a sequential looking paradigm to assess whether infants are able to differentiate synchronous from asynchronous cardiac rhythms. Twenty-nine infants (17 males, mean age = 5.13 months, SD = 0.29) viewed an animated character, moving either in synchrony or asynchrony (±10% speed) with the infant's own heartbeat, during continuous eye-tracking (*Figure 1A*). A clear visual preference for the asynchronous stimulus emerged at the group level, $M_{ASYNCH}$ = 5194 ms, SD = 2697, $M_{SYNCH}$ = 4170 ms, SD = 2167, t(28)=-3.267, p=0.0029, Cohen's d = 0.40 (*Figure 1B*, also see *Figure 1—figure supplement 1*), indicating that infants displayed an implicit sensitivity to interoceptive signals, and an ability to integrate these interoceptive signals with external visual-auditory stimuli. As with adults, there were individual differences in the direction and magnitude of preference (*Figure 1B*).

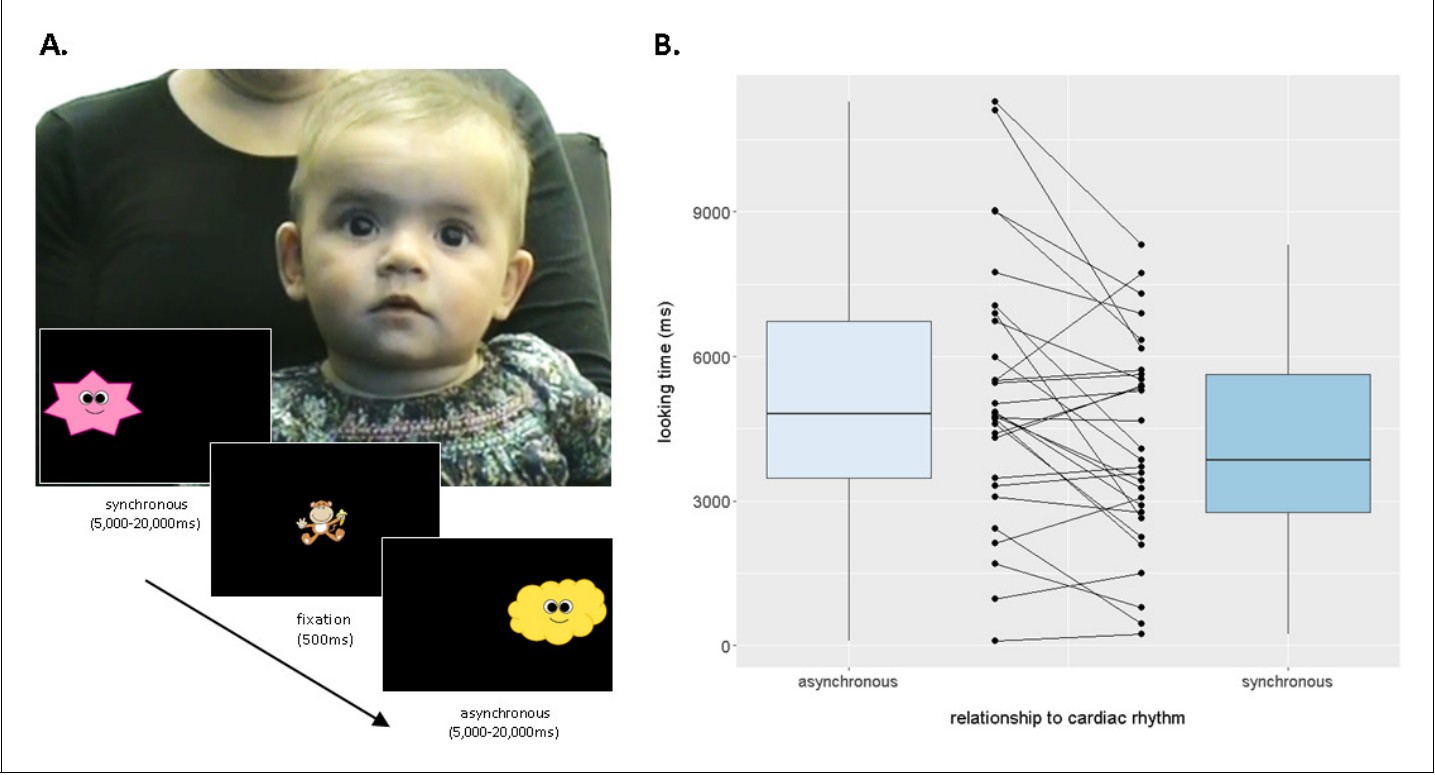

**Figure 1.** Infants differentiate between synchronous and asynchronous cardiac rhythms in a sequential looking paradigm (iBEAT task). (**A**) iBEAT paradigm. Infants viewed trials alternating between synchronous and asynchronous cardiac rhythms, presented either on the left or right of the screen. Stimuli remained on the screen for 5000 ms, after which time its continued presentation depended on infant attention (max. 20,000 ms). Informed consent was obtained from the caregiver of the infant featured in Figure 1A. (**B**) Boxplot quantification of average looking times (ms) to stimuli that were asynchronous or synchronous with infants' own cardiac rhythm (*N* = 29 infants, paired t-test, p=0.0029). Strip-chart points indicate pairs of raw data points from individual infants, reflecting individual differences in looking behaviours. Boxplot whiskers denote ±1.5*interquartile range limits.

The following source data and figure supplements are available for figure 1:

**Source data 1.** Mean looking times (ms) for individual infants for synchronous, asynchronous-faster, asynchronous-slower and (composite) asynchronous trials.

**Figure supplement 1.** Boxplot quantification of average looking times (ms) to stimuli that were −10% (slower),+10% (faster), or synchronous with infants' own cardiac rhythm (N = 29 infants, paired one-tailed t-tests, synchronous stimulus received less attention than both slower, p=0.029 and faster stimuli, p=0.0025).

**Figure supplement 2.** Boxplot quantification of average intervals (ms) between each audiovisual beat presented in the −10% (slower),+10% (faster) and synchronous trials, across the entire task.

We then measured HEP amplitude in the same infants whilst they viewed short video clips of emotional and non-emotional facial expressions (*Missana et al., 2014*). First, we investigated the relation between the behavioural cardiac discrimination performance and HEP amplitude collapsed across emotion conditions to obtain a measure of baseline HEP independent of emotion. Given the varied nature of the topography and timing of HEP in adults (*Kern et al., 2013*), we used a non-parametric Monte-Carlo cluster-based approach (*Maris and Oostenveld, 2007*) in three broad bilateral regions of interest (ROI: frontal, central and parietal, see *Figure 2—figure supplement 1*) across a 150–300 ms time window after the R wave. This time window was chosen to account for the infants' rapid heart rate (mean R-R interval = 422.98 ms, SD = 25.36). We calculated a Cardiac Discrimination score as the absolute proportion difference between looking times to the synchronous and asynchronous stimuli during the iBEAT task. Individual Monte-Carlo cluster-based regression

analyses were carried out for each of the three ROIs (*Couto et al., 2015*; *Canales-Johnson et al., 2015*). In the parietal ROI, a midline cluster (P2, POz, Pz) between 206–272 ms after the R wave was found to significantly correlate with infant cardiac discrimination, p=0.019, T$_{SUM}$ = 577.0. HEP amplitude was higher for infants who showed a greater discrimination between synchronous and asynchronous cardiac rhythms during the iBEAT task (*Figure 2*, also see Supplementary Results).

We next investigated emotion-specific modulations in HEP amplitude, by individually comparing HEPs in each emotion condition to the neutral baseline (as in [*Couto et al., 2015*]), again using a Monte-Carlo cluster analysis. In the frontal ROI, the cluster-permutation analysis identified a significant cluster for both fear observation, p=0.0015, T$_{SUM}$ = 1842.2, and for anger observation, p=0.002, T$_{SUM}$ = 1994.8. Amplitude was significantly higher during fear and anger observation than during neutral observation (Fig, 3A). The effects extended across the entire time window (150 ms-300ms after the R wave), and the clusters closely overlapped in topography (*Figure 3B*). There were no significant clusters in parietal or central ROIs, and no significant clusters emerged for the happiness vs. neutral comparison. An identical analysis on the ECG channel did not identify any significant temporal clusters discriminating between emotional conditions, confirming that the effect was specific to the *cortical* processing of the heartbeat and not due cardiac field artefacts (*Figure 3C*). An analysis on the EEG signal time-locked to the emotional stimulus, rather than the heartbeat, did not show any significant differences between emotions in the specific frontal ROI identified in the HEP

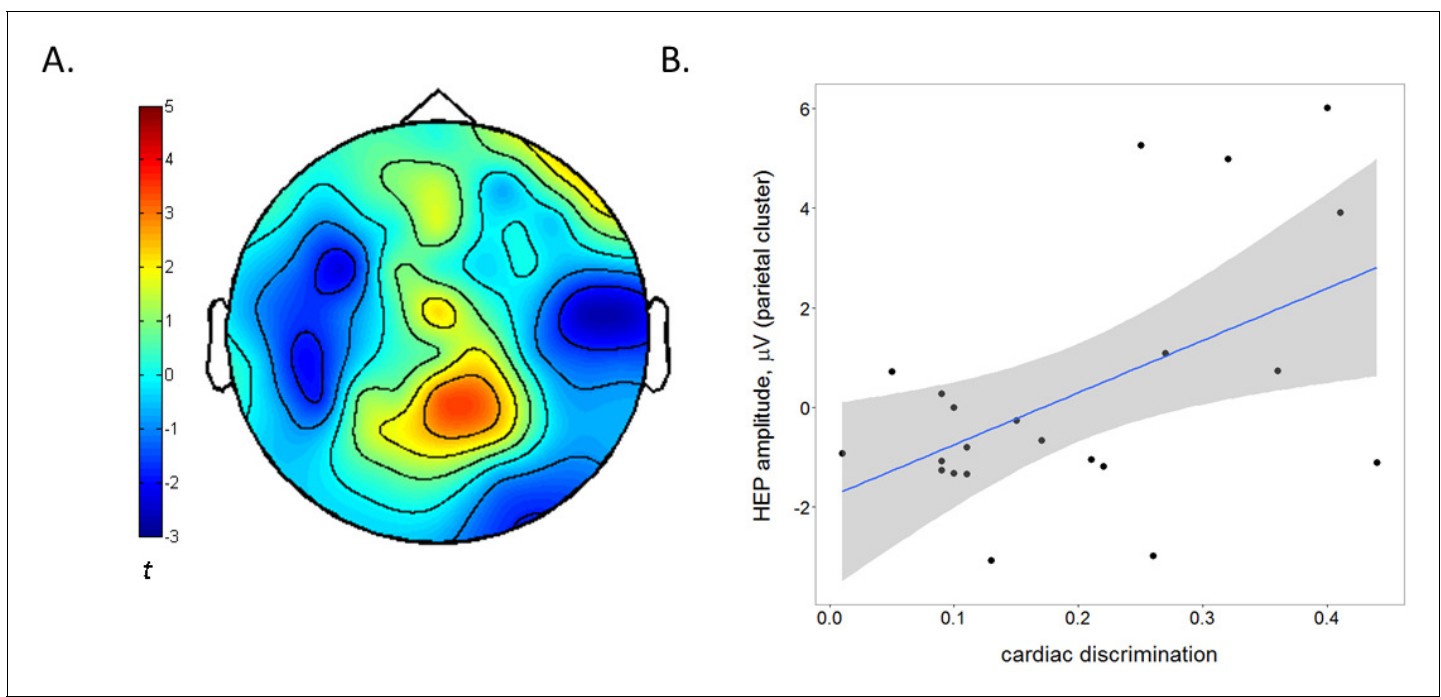

**Figure 2.** The amplitude of the Heartbeat Evoked Potential is related to individual differences in infants' behavioural cardiac discrimination. (A) Topographical representation of the significant midline parietal cluster in which HEP positively correlated with cardiac discrimination on the iBEAT task (*N* = 22 infants, Monte-Carlo cluster regression, p=0.019). Colorbar represents cluster statistic (t). (B) Scatterplot illustrating the positive correlation between cardiac discrimination and HEP amplitude. Shaded area represents 95% confidence interval of fitted regression line, *r* = 0.52, p=0.013. HEP amplitude is the subject-wise average signal from the midline parietal cluster (P2, POz and Pz) across the 206–272 ms time window of all HEP segments (irrespective of emotion observation).

The following source data and figure supplement are available for figure 2:

**Source data 1.** Cardiac discrimination scores presented for each infant, alongside mean HEP amplitude from the parietal cluster (POz, Pz, P2) in μV.

**Figure supplement 1.** Broad bilateral ROIs selected for HEP analysis; frontal, central and parietal regions, illustrated on a BioSemi 64-channel 10/20 system.

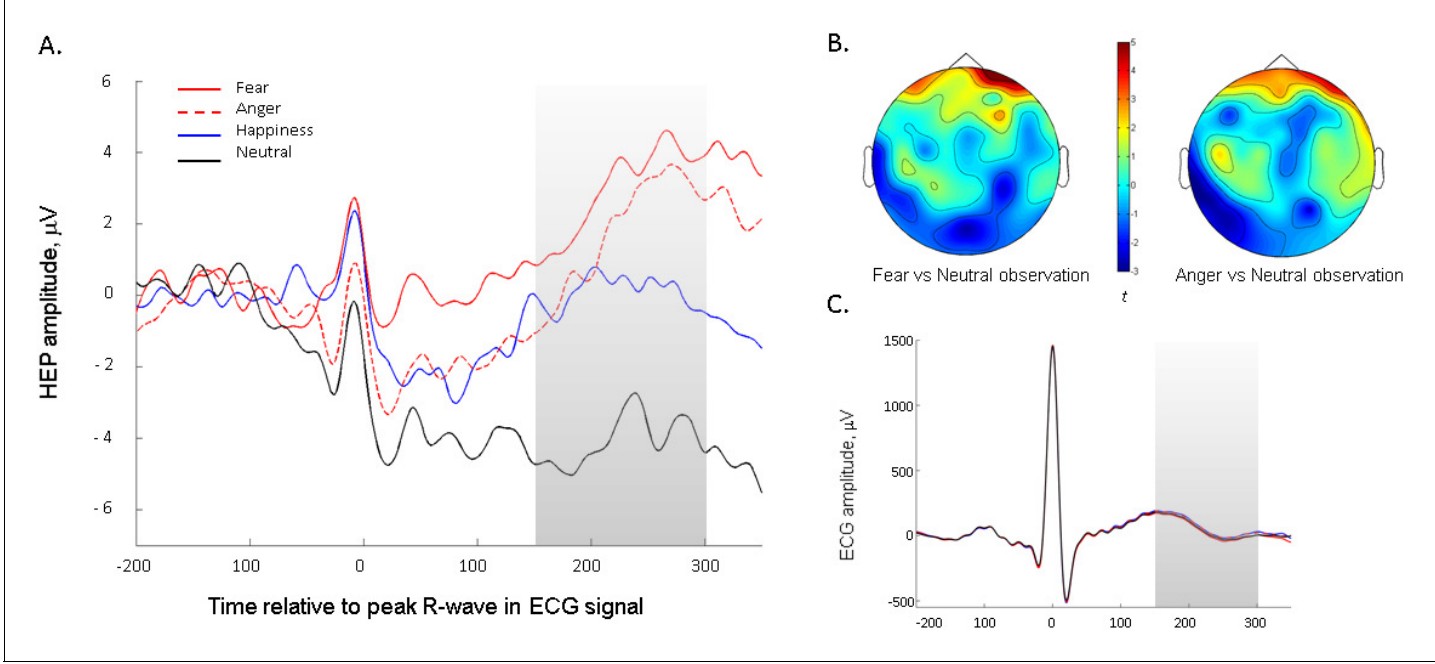

**Figure 3.** The Heartbeat Evoked Potential is modulated by infants' observation of negative emotional expressions. (**A**) Average HEP amplitude (across representative frontal channels common to fear- and anger-specific clusters) for the four emotion conditions. Shaded region represents the time-window analysed. (**B**) Topographical representation of positive frontal clusters, showing significantly higher activity during negative emotion (i.e. fear and anger) versus neutral observation (*N* = 22 infants, Monte-Carlo cluster analysis, p≤0.002). Averaged across 150–300 ms period. Colour bar shows Monte-Carlo cluster statistic (t). (**C**) Average ECG signal across the four emotion observation conditions.

analysis, suggesting that the emotional modulation of the HEP was not merely an artefact of ERPs to emotional expressions in the same areas (see Supplementary Results).

Here, we report the first evidence of implicit neurobehavioral sensitivity to interoceptive signals in early infancy. Infants' ability to discriminate cardiac-audiovisual synchrony during the iBEAT task suggests that they are not only sensitive to internal sensations, but are also able to integrate interoceptive information with stimuli in the environment to drive visual preferences. In adults, the relationship between the internal and external experience of the body has been shown to play a critical role in the malleability of body ownership (*Tsakiris et al., 2011*; *Suzuki et al., 2013*; *Aspell et al., 2013*). Therefore, the ability to integrate interoceptive and exteroceptive information may be central to infants' developing awareness of their own body boundaries. Furthermore, this integration process could be a precursor to mechanisms which allow the attribution of reward value to certain events and objects. The ability to preferentially orient to specific aspects of the environment that elicit positive interoceptive sensations, and to avoid aspects that lead to aversive or over-arousing bodily states, may provide the basis for an emerging homeostatic self-regulatory capacity in the developing infant (*Gu and FitzGerald, 2014*), via a process of interoceptive inference (*Seth, 2013*; *Allen et al., 2016*). This process is likely to be essential for guiding learning towards intrinsically motivating classes of stimuli, for example faces (*Quattrocki and Friston, 2014*). A difficulty in detecting interoceptive states, or a subsequent disruption in ascribing interoceptive 'value' to the environment, could lead to atypical development in emotional, cognitive and social domains (*Mundy and Jarrold, 2010*; *Quattrocki and Friston, 2014*). The developmental trajectory of interoception may therefore provide strong clues as to the origins of certain psychopathologies (*Quattrocki and Friston, 2014*; *Murphy et al., 2017*).

We also demonstrated that a cortical index of interoceptive processing, the HEP, is particularly sensitive to emotional processing, especially negative emotions, in infancy as it does in adulthood. These findings indicate that infants' state interoceptive processing and its cortical representation is dynamic, flexible and responsive to task demands. These results support a proposed relationship

between online cardiac signalling and emotional evaluation in early infancy, which has already received persuasive support in adults (*Garfinkel and Critchley, 2013*).

Our results also have important bearing on the study of self-awareness, the development of which has long been a topic of fascination in the behavioural sciences. It is now known that self-awareness in adulthood has close links with interoceptive abilities (*Tsakiris et al., 2011*; *Filippetti and Tsakiris, 2017*), but how or when this link emerges has never yet been investigated. Our results allow future research to assess the relationship between individual differences in interoceptive sensitivity and the achievement of various milestones in the development of self-awareness, for example mirror self-recognition or the onset of episodic remembering. The implicit, non-verbal nature of the newly-developed tasks also raise the possibility of taking a comparative approach by measuring interoception in other non-human animals (*Evrard et al., 2014*). This will provide a unique opportunity to learn more about the evolutionary trajectory of interoception and embodied self-awareness.

In conclusion, the discovery that infants are sensitive to interoceptive sensations has wide-reaching implications for the study of interoception, self-awareness and cognitive development, and paves the way for understanding how individual differences in interoception can develop, persist and influence the way in which we experience ourselves and the world.

## Supplementary results

### Supplementary results for iBEAT task

As the rhythm of the asynchronous stimuli could either be faster or slower than the infant's own heartbeat, two further one-tailed t-tests were carried out comparing these two types of asynchronous stimuli to the synchronous stimulus separately to ensure that the asynchronous preference held in both cases. Infants looked longer at both the slower stimulus, $M = 4829$, $SD = 2829$, $t(28) = -1.977$, p=0.029 (1-tailed), $d = 0.25$, and the faster stimulus, $M = 5547$, $SD = 3190$, p=0.0025 (1-tailed), $d = 0.475$, as compared to the synchronous stimulus. Looking times for the faster vs. the slower stimulus did not significantly differ, p>0.05. These analyses are illustrated in *Figure 1—figure supplement 1*.

Interbeat intervals were then calculated for each condition averaged across the group of infants; $M$(asynch-faster) = 375 ms ($SD$ = 15), range = 361–392 ms, $M$(asynch-slower) = 417 ms ($SD$ = 18), range = 399–436 ms, $M$(synch) = 398 ms ($SD$ = 18), range = 376–419 ms. These data are displayed in *Figure 1—figure supplement 2*.

### Supplementary results for HEP

To ensure that the correlation found between cardiac discrimination and HEP amplitude was not due to cardiac field artefacts, we also calculated the correlation coefficient between the cardiac discrimination score and the average ECG amplitude across the time window of significance (206–272 ms). The correlation was non-significant, $r = -0.15$, p=0.502, suggesting that the observed relationship between cardiac discrimination and HEP amplitude was specific to the cortical processing of the heartbeat rather than cardiac activity per se.

To ensure that the emotional modulation of HEP observed for fear and anger was not instead caused by an ERP to the emotional expressions in the same channels, we also carried out an analysis of the EEG signal time-locked to the onset of emotional expressions, instead of heartbeats. Segmentation was performed surrounding the entire duration of each emotional expression (2000 ms). We employed the same Monte-Carlo random cluster permutation method as used for the HEP analysis, to ensure comparability. First, the entire time-window was interrogated for spatiotemporal clusters that significantly discriminated fear or anger from neutral conditions, within the eleven channels that we found emotional modulation of HEP (see *Figure 2—figure supplement 1*). No significant clusters were revealed in these channels, neither for the anger-neutral comparison (p≥0.213) nor the fear-neutral comparison (p≥0.395). We repeated this analysis with a more specific temporal focus on 300–600 ms after emotional expression onset, which has been identified as a likely time-window for infant ERP to emotional expressions (*Kobiella et al., 2008*). Again, no significant clusters were revealed, either for the fear-neutral comparison (no clusters) or the anger-neutral comparison (p=0.279). Finally, we extracted average amplitude of the EEG signal from the critical channels time-locked to the emotional expression for each individual infant, and calculated correlation coefficients between this signal and the average HEP amplitude during observation of that same emotion. If our

HEP results were actually caused by ERP signal fluctuations, there should be a significant correlation between these two variables. Importantly, there were no significant correlations between average ERP amplitude and HEP amplitude, either for fear ($r$ = –0.171, p=0.470) or anger ($r$ = 0.175, p=0.436). The results of these analyses suggest that the emotional modulation of the HEP signal in frontal areas is not an artefact of more general ERPs in response to these emotions in the same locations.

## Materials and methods

### Participants

Forty-one healthy, full-term infants were tested in total, at 5 months of age (19 males, mean age = 5.10 months, SD = 0.29). The expected effect sizes were not known in advance, so samples were selected according to similar adult literature (e.g. [*Fukushima et al., 2011*]) assuming an approximate 50% attrition rate, which is usual in infant EEG studies with this age range. Infants were recruited using a marketing company database, which provides data information from consenting mothers to be. Recruitment leaflets were sent to each household. Parents were able to participate by signing up to our online database or by contacting us via email. The study was completed in one session, and conducted according to the Declaration of Helsinki and all methods were approved by the Royal Holloway University of London Departmental Ethics Committee.

### Measures
#### iBEAT task

For this task, infants were seated in a high-chair approximately 60 cm away from a computer screen integrated into a Tobii T120 eyetracker. Three disposable paediatric AgCl electrodes were attached to the infants' chest and stomach in a 3-lead ECG setup, to monitor their cardiac activity throughout the task (Powerlab, ADInstruments, www.adinstruments.com). The R-peaks of the ECG were identified online using a hardware-based detection function ('Fast Output Response' function) from ADInstruments, and stimulus presentation was managed by a custom-made algorithm implemented in Matlab 2015a (MathWorks, Natick, MA). The onset of each R wave in the ECG trace was defined as the moment the voltage exceeded a predefined threshold, set individually for each infant. Once the function detected a R wave, a transistor-transistor logic (TTL) pulse was then sent to the computer presenting the audiovisual stimuli, with a delay of less than 2 ms as confirmed by internal lab reports.

In each trial, infants were shown an animated character, which moved rhythmically up and down, either synchronously with the infant's own heartbeat, or asynchronously (±10% speed). Each up-down movement was accompanied by an attractive sound to mark the rhythm. For synchronous trials, the custom-made algorithm presented an auditory tone and change in position of the animated character upon the receipt of every TTL pulse. For asynchronous trials, the algorithm produced a cardiac-like rhythm that was ±10% the speed of the infant's average heart rate recorded from the previous trial. Trials alternated between showing either asynchronous or synchronous movement, and the character could appear either on the left or the right of the screen (see [*Phillips-Silver and Trainor, 2005*]).

Once a steady ECG trace had been achieved, and the eyetracker had been calibrated to accurately detect the infant's fixations, the computer task was started. Infants were shown the character for a minimum of 5 s, after which the continued presentation of the character was contingent on the infant's attention. If the infant continued to look at the character, it remained on the screen for a maximum of 20 s. If the infant looked away from the character for longer than three consecutive seconds, the trial was automatically terminated by the computer program and the next trial begun. Between each trial, an attractive central fixation clip was presented to re-engage the infants' attention. The task was terminated when four consecutive trials did not receive sufficient fixation to extend the trial time past the five second minimum, or when the infant became too tired or fussy to continue.

## HEP measurement

The EEG recording was prepared whilst the infant was seated on their caregiver's lap. A BioSemi elastic electrode cap was fitted onto the infant's head, into which 64 active Ag-AgCl electrodes were fixed according to the 10/20 system. An external electrode was attached to the infant's upper left abdomen to provide an ECG trace sufficient to detect the R-wave offline. The EEG and ECG signals were recorded using the BioSemi ActiveTwo system (BioSemi, Amsterdam, The Netherlands), with a sampling rate of 1024 Hz and band-pass filtered online at 0.16–100 Hz. The standard BioSemi references Common Mode Sense (CMS) electrode and Driven Right Leg (DRL) electrode were used.

Once clear EEG and ECG signals were obtained, the infant was repositioned 60 cm in front of a 29' computer screen. The caregiver was asked to avoid interacting with the infant or looking at the screen during the task. The task comprised of short video clips of females making emotional expressions, following the paradigm of *Missana et al. (2014)*. The standardised clips were sourced from the Montréal Pain and Affective Face Clips (MPAFC [*Simon et al., 2008*], RRID:SCR_015497) database. Happiness, fear, and anger were selected as emotions of interest. A neutral stimulus was also included, in order to provide a 'no emotion' control condition for comparison. Each video was modified to last for three seconds in total; this duration was comprised of a 1 s period showing the face with a neutral expression, then a 1 s dynamic period when the emotional expression manifested, and a final second freeze-frame of the expression at its peak intensity (*Missana et al., 2014*). The videos featured four different female models, and were cropped around the models' faces using a soft circular template to remove extraneous features.

Trials were presented in a pseudo-randomized order, ensuring that trials featuring the same model and emotion were not presented consecutively. Between each trial, infants were shown an attention-getter, consisting of a short audio-visual clip of an engaging, non-social object presented centrally on the screen. When infants' attention failed to return to the screen for two consecutive trials, the experimenter (positioned behind a curtain partition) initiated a longer video clip with music which gave the infant a short break and an incentive to reorient to the screen. Once the infant's attention was recaptured, the experimenter manually recommenced the next trial. If the infant showed prolonged inattention (defined as having not attended to four consecutive trials) or fussiness, the task was terminated. A maximum of 48 trials were presented. Stimulus presentation was controlled using Presentation software (Neurobehavioural Systems, Inc.).

## Testing procedure

Mothers and infants familiarised themselves with the testing room. Infants first performed the iBEAT task. After a short break for feeding and changing if needed, they were then placed on their mother's lap for the EEG cap and electrodes to be fitted. Once the EEG signal was clear and the infant was ready, they were returned to the high-chair and the Emotion Observation Task was run. The cap and ECG electrodes were then removed, and mother and infant were taken to a comfortable rest area to take a break, feed and change if necessary. The mother then completed a brief questionnaire and further behavioural task (the results of which will be reported in a separate publication) before being thanked, debriefed and given a small gift and monetary compensation for travel costs.

## Data pre-processing

### iBEAT task

Looking time data were processed offline. Only trials in which there were no movement artefacts in the ECG recording were included in the analysis. Infants were excluded if they did not complete a minimum of eight trials in total (four in the synchronous and four in the asynchronous conditions). In total, seven infants were unable to commence the task, due to the eyetracker being unable to detect the infant's eyes (N = 5) or the ECG signal having a high level of interference (N = 2). Of the remaining 36 infants who took part in the task, seven infants were excluded due to insufficient artefact-free trials (19%), leaving 29 infants remaining in the final analysis (see *Table 1*).

Total looking time in each trial was calculated as the summed duration of all recorded eyetracker samples falling in the region encompassing the location of the animated character during that trial. Outliers in the looking time data, defined as any trial which received an individual looking time greater than two standard deviations away from the group mean for that condition, were then removed from the sample. For the purpose of correlating heartbeat discrimination ability with HEP

**Table 1.** Mean number of valid, artefact-free trials in the iBEAT task included in the final sample, for each age group and condition. Standard deviation from the mean indicated in brackets. This table relates to the data displayed in *Figure 1*.

**Average number of trials completed (SD)**

| Synchronous | Asynch - slower | Asynch - faster | Total |
|---|---|---|---|
| 7.18 (2.96) | 4.04 (1.62) | 3.82 (1.47) | 15.04 (4.87) |

amplitude, a discrimination score was calculated reflecting the absolute proportion difference between looking times to the synchronous and asynchronous stimuli. An absolute preference was considered important as it reflected an ability to detect cardio-visual synchrony, regardless of whether the infant had a preference towards or against it. This is because there are high inter-individual and intra-individual differences in whether an infant will show a familiarity or a novelty preference, particularly in intermodal looking time procedures (*Houston-Price and Nakai, 2004*)

## HEP measure

Out of the full sample, thirty-nine infants completed the EEG recording. Offline EEG pre-processing was performed using BrainVision Analyzer software (Brain Products, Munich, Germany). The continuous EEG data was filtered offline with a bandpass filter of 0.1–30 Hz (24 dB/oct) and a 50 Hz notch filter. An Independent Component Analysis (ICA) was performed on raw EEG data to remove eye-movement artefacts and Cardiac Field Artefacts (CFA) (*Terhaar et al., 2012*). Using ICA to remove CFAs has been shown to be highly effective in removing cardiac artefacts from the EEG signal (*Terhaar et al., 2012*; *Park et al., 2014*; *Luft and Bhattacharya, 2015*). Bad channels were replaced using topographic interpolation.

The data was then segmented into a 2000 ms epoch (1000 ms to 3000 ms) time-locked to the onset of the presentation of the emotion stimulus. This period was selected as this was when the observed emotional expression was at peak intensity. The data was then re-segmented to extract the HEPS, with a duration of 600 ms (−200 ms to 400 ms) time-locked to the R-wave. The resulting segments were baseline corrected using an interval from −200 ms to −50 ms to avoid artefacts from the R wave rising edge (*Canales-Johnson et al., 2015*). Semi-automatic artefact rejection was combined with visual inspection for all participants. Epochs exceeding a voltage step of 200 µv/200 ms, a maximal allowed difference of 250 µv/200 ms, amplitudes exceeding ±250 µV, and low activity less than −0.5 µV/50 ms were rejected from analyses. Infants who completed less than 25% of the total trials (less than five trials) in each emotion condition were automatically excluded. In total, 7 infants (18%) of the infants were excluded from the EEG analyses based on these criteria which is in line with other infant EEG studies. Segments were then re-referenced to the average and grand averaged (see *Table 2*).

As a central aim was to investigate whether the observed behavioural cardiac discrimination was related to HEP amplitude during the EEG recording, we restricted EEG analysis to those infants who had valid, complete data from both the EEG task and the task (N = 22). To investigate whether behavioural cardiac discrimination was related to HEP amplitude, HEP segments were collapsed across emotion conditions in order to get an estimate of HEP amplitude independent of specific emotional processing demands.

**Table 2.** Mean number of trials, heartbeats and valid HEP segments extracted for each emotion condition (standard deviation in brackets), N = 22. This table relates to data displayed in *Figure 3*.

| Condition: | Happiness | Fear | Anger | Neutral |
|---|---|---|---|---|
| Average Number of Trials | 12.5 (3.6) | 12.4 (3.6) | 12.7 (3.7) | 12.7 (3.6) |
| Average Number of Heartbeats | 46.6 (12.6) | 45.8 (12.3) | 46.3 (12.7) | 47.4 (12.8) |
| Average Number of HEP segments | 22.0 (13.9) | 21.3 (13.9) | 23.8 (13.9) | 21.8 (15.0) |

## Statistics

All tests were evaluated against a two-tailed p<0.05 level of significance. For the HEP analysis, a Monte-Carlo random cluster-permutation method was implemented in FieldTrip. This method corrects for multiple comparisons in space and time (*Maris and Oostenveld, 2007*). Using this method, all samples that showed a significant (p<.05) relationship with our independent variable were clustered according to spatiotemporal adjacencies, and cluster-level statistics were calculated by taking a sum of the t-values for each cluster. A Monte-Carlo permutation method then generated a p-value by calculating the probability that this cluster-level statistic could be achieved by chance, by randomly shuffling and resampling the independent variable structure a large number of times (2000 repetitions) (*Maris and Oostenveld, 2007*). Spatiotemporal clusters that had a resulting Monte-Carlo corrected *p*-value of less than the critical alpha level of. 05 were interpreted as 'significant'.

For both the iBEAT task and the HEP measurement, data collection was not performed blind to the experimental condition to which each trial belonged, due to the requirements for stimulus and cardiac monitoring during the task. However, for HEP analysis, experimental condition was removed from the data after data collection and all EEG pre-processing was performed blind to the conditions of the experiment. Condition was revealed at final statistical analysis so that specific emotions could be compared to the neutral condition. Both reported tasks had within-subjects designs involving no group allocation; therefore, blinding to any between-subject conditions and randomization to such conditions was not applicable.

## Acknowledgements

This study was supported by a BIAL Foundation Grant for Research Project 087/2014 to LM and MT and the European Research Council [ERC- 2010-StG-262853] under the FP7 to MT.

## Additional information

### Funding

| Funder | Grant reference number | Author |
|---|---|---|
| Fundação Bial | 87/14 | Lara Maister<br>Manos Tsakiris |
| H2020 European Research Council | ERC- 2010-StG-262853 | Manos Tsakiris |

The funders had no role in study design, data collection and interpretation, or the decision to submit the work for publication.

### Author contributions

LM, Conceptualization, Data curation, Formal analysis, Supervision, Funding acquisition, Methodology, Writing—original draft, Writing—review and editing; TT, Data curation, Investigation, Methodology; MT, Conceptualization, Supervision, Funding acquisition, Writing—review and editing

### Author ORCIDs

Lara Maister, http://orcid.org/0000-0001-8308-9722

### Ethics

Human subjects: Informed consent was obtained from the caregivers of all infants involved in the study. Informed consent was also obtained from the caregiver of the infant featured in Figure 1A. All methods were explicitly approved by the Royal Holloway Department of Psychology Ethics Committee, reference code 2015/050R1.

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
