## [Decision Letter]

Thank you for submitting your article "Neurobehavioral evidence of interoceptive sensitivity in early infancy" for consideration by *eLife*. Your article has been reviewed by two peer reviewers, and the evaluation has been overseen by Sid Kouider acting as the Reviewing Editor and Sabine Kastner as the Senior Editor. The following individual involved in review of your submission has agreed to reveal her identity: Sarah Garfinkel (Reviewer #2).

The reviewers have discussed the reviews with one another and the Reviewing Editor (Sid Kouider) has drafted this decision to help you prepare a revised submission.

Summary:

The reviewers found that your paper describes a novel and interesting study investigating interoceptive sensitivity in infants, but a number of important substantive points have been raised. In particular, the study remains inconclusive as to whether the neural signature used in your study really taps onto the modulation of interoception by emotional stimuli. In addition, the reviewers and editors believe that the methods should be clarified as they remained difficult to grasp for assessing the validity of your findings. Further analysis are also required to account for potentials biases in the results (i.e., the potential effect of novelty surrounding trial type, see comments by reviewer 2), and some of the interpretations appears to be too speculative and should probably be toned down (see comments by reviewer 1). In light of these issues, we cannot offer to publish your paper in *eLife*, as it stands. If, however, you feel that you can address all of the points raised below, and make a strong case, we shall be willing to consider a revised version of your paper for publication.

Essential revisions:

*Reviewing Editor:*

A major issue concerns the interpretation derived from the fact that emotional expressions modulate Heartbeat Evoked Potentials (HEP, Figure 3). The HEP is an EEG component, which is time-locked to the heart beat. However, although they are differences across emotions, this might be the reflection of other sources that would show processing difference regardless of the heart beat: they might reflect the modulation of event-related potentials triggered by the emotional stimuli, although these amplitude differences would still show up when time-locking the analysis to the heart beat. In other terms, the reader might find this finding inconclusive and it seems important to show some degree of specificity to the HEP. This might be done by inspecting the results time-locked to the stimulus onset rather than the HEP, which might differ, or to show that some aspects of the HEP (for instance the phase) are 'specifically' modulated by emotional stimuli. This might also be discussed in an attempt to make the interpretation of your results more convincing and less ambiguous.

*Reviewer 1:*

Regarding the same results depicted on Figure 3, the difference in topography of this finding and of the areas involved in the correlation with the behavioural measure also suggests that they may not both reflect interception. It might be important to look at the onset of the heart beat because one would expect that if the difference really reflects interception, it should not depend on when after the face onset the beats appear. Though, the face presentations were very short, and it is not clear whether this would leave a lot of scope for testing this, and the differences in response to emotional faces could be sustained. And it is possible that the effects on interception are stronger closer to the onset of an emotional stimulus.

The methodology is not sufficiently described to assess the validity of findings.

How was the visual stimulation synchronised to the infant's heart beat? How did you achieve +/- 10% speed?

In most infant looking studies a cut-off of 2 seconds is used to determine when attention was terminated. Justify why 3 seconds were used here.

The ECG channel activity should also be correlated with the cardiac discrimination score, to rule out a cardiac contribution to the association between HEP and cardiac discrimination. This is justified also by the choice of a rather early time window for analysis – 150 ms after the R wave (compared to 50 ms after the T wave in Park et al., 2014) and which might correspond to a section of the HEP artefacted by the cardiac response itself.

Was the heart beat latency from the onset of the visual stimuli, similar across conditions? A systematic difference in latency could lead to differential effects that do not reflect emotional processing but more general arousal/orienting.

Discussion should stay closer to what the current data can tell about infant cognition or better justify the potential broader implications.

The Discussion is rather speculative and the inferences made from the data unsubstantiated. The authors only measured discrimination between synchronous and asynchronous heart rate. Although there is a tendency to prefer asynchronous stimuli, this is not sufficient to infer that infants assigning positive or aversive value to this perception.

As the authors correctly state, the HEP findings suggest that facial expressions influence the processing of interoceptive signals. The following (reversed) inference: "Even at this early age, infants' response to emotions appear to be closely influenced by their visceral reactions." cannot be drawn from the current data.

The link to self-awareness seems un-motivated. Please explain why interception rather than any bodily sensation, is relevant to self-awareness.

*Reviewer 2:*

Trial number: I understand that the authors wanted to balance the amount of synchronous and asynchronous trials, and this is correct. However, because there are two asynchronous conditions this has led to double the amount of trials for the synchronous condition relative to each of the asynchronous conditions (faster, slower). Thus, the effect of novelty surrounding trial type also needs to be controlled for. Within the current paradigm there are three classes of stimulus speeds: average, fast, slow, that map onto synchronous, asynchronous and asynchronous. As there is reduced looking time for the synchronous trials, and there are also double these trials relative to fast and slow speeds, it is necessary to show that this difference does not derive from a reduction in novelty simply due to increased instances of this (average) trial speed. Could the authors please repeat analyses but using only the first half of synchronous trials to demonstrate that the direction of looking time still holds? It is ok if differences are now not significant, as the reduction in power from slashing half the trials is likely to be considerable, but it would be helpful to see that the direction of the effect is still present within the data (and that reduced looking time is not driven by the last half of synchronous trials). This type of analysis could be problematic if it then means more early trials for synchronous relative to asynchronous trials are included (and, say, looking time is significantly reduced with time). Thus the authors should feel free to offer their own solution on how to remove the confound between trial number, speed and synchronicity.

[Editors' note: further revisions were requested prior to acceptance, as described below.]

Thank you for resubmitting your work entitled "Neurobehavioral evidence of interoceptive sensitivity in early infancy" for further consideration at *eLife*.

After reading your reply to reviewers and your revised manuscript, I found your revision satisfying. You addressed all the points raised by the reviewers and myself, and the extra analysis you report seems to contradict the major issue I raised about the possibility that you might be measuring ERPs rather than HEP to emotional pictures. I am thus willing to accept your paper for publication, but I would like to point to one remaining issue that needs to be addressed before full acceptance.

The extra analysis you carried out to disentangle ERP from HEP is reported in the reply while this should appear in manuscript, as it is not about just convincing me or the reviewers but also the reader that might have the same issue in mind when reading your paper. This could be done briefly in the Results or Discussion sections and with further details in the supplementary material.

---

## [Author Response]

*Essential revisions:*

*Reviewing Editor:*

*A major issue concerns the interpretation derived from the fact that emotional expressions modulate Heartbeat Evoked Potentials (HEP, Figure 3). The HEP is an EEG component which is time-locked to the heart beat. However, although they are differences across emotions, this might be the reflection of other sources that would show processing difference regardless of the heart beat: they might reflect the modulation of event-related potentials triggered by the emotional stimuli, although these amplitude differences would still show up when time-locking the analysis to the heart beat. In other terms, the reader might find this finding inconclusive and it seems important to show some degree of specificity to the HEP. This might be done by inspecting the results time-locked to the stimulus onset rather than the HEP, which might differ, or to show that some aspects of the HEP (for instance the phase) are 'specifically' modulated by emotional stimuli. This might also be discussed in an attempt to make the interpretation of your results more convincing and less ambiguous.*

We have now carried out an ERP analysis to investigate whether the emotional modulation of HEP that we report is distinct from the more general emotional modulation of ERPs. We accordingly entered the data from all infants included in the HEP analysis into our ERP analysis in order to detect any overlap in topography, which may cast doubt over our initial findings. The analysis was carried out using a Monte-Carlo random cluster permutation method, with a cluster-corrected p-value threshold of. 05, identical to the one used throughout the manuscript to ensure comparability.

When interrogating the EEG signal for spatiotemporal clusters differentiating ERPs to fear or anger from neutral trials (the key contrasts for our HEP findings) across the entire duration of the emotional stimulus, no significant clusters were revealed in the frontal channels that the HEP analysis had identified as sensitive to fear and anger, respectively (hitherto referred to as the key ROIs). We then calculated the average amplitude from our ERP data in the key ROIs, and tested whether it correlated with average HEP amplitude; this should be the case if our HEP results were actually caused by ERP signal fluctuations. However, these correlations were non-significant for both fear (r=-.011, p=.962) and anger (r=.279, p=.209). Finally, we carried out a more specific analysis focussing in on the time period from 300-600ms after emotional expression onset, which has identified as a likely time-window of infant ERP to emotional expressions (following Kobiella, Grossmann, Reid & Striano, 2007). Again, no clusters were identified that were significantly modulated by emotion in this time window in the key ROIs, and there were no significant correlations between average amplitude and HEP amplitude, either for fear (r-.171, p=.470) or anger (r=.175, p=.436). The results of these analyses suggest that the emotional modulation of the HEP signal in frontal areas is not an artefact of more general ERPs in response to these emotions in the same locations. At the same time we would like to reiterate that our experimental paradigm was designed to enable us to analyse HEP during emotional observation (of which there are many occurring in each emotion trial), As a result the infants included in our analysis did not necessarily complete a sufficient number of artefact-free emotion trials to provide as a reliable and detailed ERP analysis (in which only one segment can be obtained from each trial as one would expect for an experimental design tailored to study ERPs to emotional faces, e.g. Missana, Grigutsch & Grossmann, 2014).

*Reviewer 1:*

*Regarding the same results depicted on Figure 3, the difference in topography of this finding and of the areas involved in the correlation with the behavioural measure also suggests that they may not both reflect interception. It might be important to look at the onset of the heart beat because one would expect that if the difference really reflects interception, it should not depend on when after the face onset the beats appear. Though, the face presentations were very short, and it is not clear whether this would leave a lot of scope for testing this, and the differences in response to emotional faces could be sustained. And it is possible that the effects on interception are stronger closer to the onset of an emotional stimulus.*

Although we agree that it would be interesting to investigate how the emotional modulation of HEP changes across the time course of a trial, we fully agree with the several valid challenges that the reviewer raises to this approach, including the issue of our short trials which are not optimised for an analysis of the effects of stimulus duration. Furthermore, it is not clear what would be predicted from such an analysis, or what results would strengthen or challenge our existing conclusions. It is certainly possible that the emotional modulation of HEP is strongest at the onset of an emotional expression, but conversely it is also feasible that the emotional modulation increases as the intensity of the emotional expression builds, and we do not think that either of these plausible options preclude the involvement of interoception. Therefore, on the basis of the reasons provided by reviewer 1 and the nature of our experimental design, we agree that this analysis wouldn’t be informative. Perhaps a different experimental design that has shortened or even static facial expressions could perhaps answer the reviewer’s question, and will be considered in future experiments.

*The methodology is not sufficiently described to assess the validity of findings.*

*How was the visual stimulation synchronised to the infant's heart beat? How did you achieve +/- 10% speed?*

The following details have been added to the manuscript:

*“*The R-peaks of the ECG were identified online using a hardware-based detection function (“Fast Output Response” function) from ADInstruments, and stimulus presentation was managed by a custom-made algorithm implemented in Matlab 2015a. […] Trials alternated between showing either asynchronous or synchronous movement, and the character could appear either on the left or the right of the screen.*”*

*In most infant looking studies a cut-off of 2 seconds is used to determine when attention was terminated. Justify why 3 seconds were used here.*

Although 2 seconds is a common cut-off in infant looking paradigms, previous infant studies have varied and several have used either 1 second (e.g. Phillips-Silver & Trainor, 2005) or 3 seconds, (e.g. Mareschal & Johnson, 2003). We decided to use the longer cut-off of 3 seconds because during piloting, we observed that infants often became very active with accompanying head and limb movement during the rhythmic stimulus which could have resulted in temporary loss of eye-tracking signal. Given that our stimulus presentation was automatically controlled by an online algorithm based on the eyetracking signal, rather than experimenter-controlled, we aimed to minimise the number of valid trials terminated prematurely.

*The ECG channel activity should also be correlated with the cardiac discrimination score, to rule out a cardiac contribution to the association between HEP and cardiac discrimination.*

Thank you for this important point. The following passage has been added to the manuscript:

“To ensure that the correlation found between cardiac discrimination and HEP amplitude was not due cardiac field artefacts, we then calculated the correlation coefficient between the cardiac discrimination score and the average ECG amplitude across the time window of significance (206-272ms). The correlation was non-significant, r=-.15, p=.502, suggesting that the observed relationship between cardiac discrimination and HEP amplitude was specific to the cortical processing of the heartbeat rather than cardiac activity per se.”

*This is justified also by the choice of a rather early time window for analysis – 150 ms after the R wave (compared to 50 ms after the T wave in Park et al., 2014) and which might correspond to a section of the HEP artefacted by the cardiac response itself.*

The earlier time-window selected for analysis was necessary due to the very rapid nature of the infant heart rate. This is explained in the fifth paragraph of the main text. The Park et al. study referred to is an MEG study rather than EEG, and most crucially uses adult participants whose heart rates are approximately half that of our infants. Although ECG components other than the R wave are not easy to identify from our data due to the montage of the external electrode, it is likely that the 150-300ms post-R-wave window we selected does indeed encompass the T wave of the infant ECG. Our 300ms upper cut-off ensures that the time-window was not contaminated by the following heartbeat.

*Was the heart beat latency from the onset of the visual stimuli, similar across conditions? A systematic difference in latency could lead to differential effects that do not reflect emotional processing but more general arousal/orienting.*

This is an interesting point, but as you know it is theoretically difficult to separate arousal from emotion (see e.g. Luft, C. D. B., & Bhattacharya, J. (2015). Aroused with heart: modulation of heartbeat evoked potential by arousal induction and its oscillatory correlates. Scientific Reports, 5, 15717). Therefore, from a conceptual point of view, we would not want to claim that our results were attributed to ‘pure’ emotional processing devoid of changes in arousal. However, there are also some methodological challenges raised by an attempt to analyse heartbeat latency as the reviewer suggests. We used dynamic emotional facial expressions in our study partly to avoid the potentially confounding role of an orienting response to the sudden onset of a full-intensity static emotional expression. As these emotional stimuli were videos, which initially showed a neutral expression before dynamically moving into showing an emotion, it is unclear to us what the ‘onset’ of the emotional expression could be defined as, i.e. the moment that infants would be predicted to have a cardiac orienting/attentional response.

However, to allay any concern, we analysed the latency between the onset of the dynamic period of the emotional video and the infants’ next heartbeat, for all four emotion conditions. We then compared the latency for fear and anger conditions to the neutral condition, to mirror the key significant comparisons yielding our HEP results. The latencies for both fear (M=207ms, SD=58) and anger (M=238, SD=56) did not significantly differ from those of the neutral condition (M=221, SD=59, p>.05). Therefore, it seems unlikely that systematic differences in heartbeat latency can explain our HEP results.

*Discussion should stay closer to what the current data can tell about infant cognition or better justify the potential broader implications.*

*The Discussion is rather speculative and the inferences made from the data unsubstantiated. The authors only measured discrimination between synchronous and asynchronous heart rate. Although there is a tendency to prefer asynchronous stimuli, this is not sufficient to infer that infants assigning positive or aversive value to this perception.*

Thank you for this point. We did not intend to claim that our data proved that infants assigned positive or negative value to interoceptive signals, but instead wanted to suggest that a mechanism by which interoceptive information is integrated with exteroceptive stimuli could be a possible means by which positive or negative interoceptive value is assigned to stimuli in the environment. We have now reworded the relevant section to suggest that the integration process may be a *precursor* to the development of mechanisms which allow the attribution of reward value to certain events and objects (rather than sufficient for that attribution, as we inadvertently suggested in our previous draft): “In adults, the relationship between the internal and external experience of the body has been shown to play a critical role in the malleability of body ownership20–22. […] Furthermore, this integration process could be a precursor to mechanisms which allow the attribution of reward value to certain events and objects.”

*As the authors correctly state, the HEP findings suggest that facial expressions influence the processing of interoceptive signals. The following (reversed) inference: "Even at this early age, infants' response to emotions appear to be closely influenced by their visceral reactions." cannot be drawn from the current data.*

We have now removed this statement, as we agree that the reversed inference is not valid.

*The link to self-awareness seems un-motivated. Please explain why interception rather than any bodily sensation, is relevant to self-awareness.*

There is a wide and rapidly-growing literature specifically linking interoception with self-awareness. We do not claim that only interoceptive sensations are relevant for self-awareness, but instead we emphasize the role that interoceptive information together with exteroceptive information about the body in co-constituting self-awareness, as other highly-cited theoretical accounts have suggested (e.g. Damasio, 2010; Craig, 2009), and have recently received growing empirical support (e.g. Tsakiris, Tajadura-Jiménez & Costantini, 2011; Suzuki, Garfinkel, Critchley, & Seth, 2013; Aspell, Lenggenhager, & Blanke, 2009; Allen et al., 2016; Seth, 2013). Readers are directed to some of these references, in the seventh paragraph of the main text, to back up our point.

*Reviewer 2:*

*Trial number: I understand that the authors wanted to balance the amount of synchronous and asynchronous trials, and this is correct. However, because there are two asynchronous conditions this has led to double the amount of trials for the synchronous condition relative to each of the asynchronous conditions (faster, slower). Thus, the effect of novelty surrounding trial type also needs to be controlled for. Within the current paradigm there are three classes of stimulus speeds: average, fast, slow, that map onto synchronous, asynchronous and asynchronous. As there is reduced looking time for the synchronous trials, and there are also double these trials relative to fast and slow speeds, it is necessary to show that this difference does not derive from a reduction in novelty simply due to increased instances of this (average) trial speed. Could the authors please repeat analyses but using only the first half of synchronous trials to demonstrate that the direction of looking time still holds? It is ok if differences are now not significant, as the reduction in power from slashing half the trials is likely to be considerable, but it would be helpful to see that the direction of the effect is still present within the data (and that reduced looking time is not driven by the last half of synchronous trials). This type of analysis could be problematic if it then means more early trials for synchronous relative to asynchronous trials are included (and, say, looking time is significantly reduced with time). Thus the authors should feel free to offer their own solution on how to remove the confound between trial number, speed and synchronicity.*

We thank the reviewer for this important point. It is correct that infants saw twice as many trials classified as ‘synchronous’ than those classified as ‘faster’ or ‘slower’. However, crucially, the rate of presented beats for the asynchronous trials was calculated as ± 10% of the mean heart rate of the previous (synchronous) trial. We apologise that we had omitted these key details regarding how the rhythm of the asynchronous trials were presented in the original manuscript (also see reviewer 1 point 2) and so we have now inserted more information in the Materials and methods section.

Due to normal fluctuations in infants’ heartrate during the task, our method meant that individual trials within a certain trial type differed with regards to their speed of beat presentation. For example, any given ‘asynch-faster’ trial was 10% faster relative to the infant’s heart rate during the previous trial, and therefore was unlikely to be exactly the same rhythm as any other trial classified as ‘asynch-faster’. As infants’ heart rates fluctuated, any given trial categorised as, for example, ‘asynch-faster’ could easily be presented at a rhythm close to, or slower than, a randomly selected trial categorised as ‘synchronous’, and the same for ‘asynch-slower’ trials. Thus, the trial designations (synch, asynch-faster, asynch-slower) are only relevant in the context of the infant’s changing heartbeat rhythm and do not reflect absolute or fixed stimulus characteristics relative to an external measure (i.e. fixed number of milliseconds between beats).

To illustrate this, we calculated the mean, standard deviation and range of inter-beat intervals for each infant and each condition; M(asynch-faster) = 375ms (SD=15), range = 361-392ms, M(asynch-slower) = 417ms (SD = 18), range = 399-436ms, M(synch) = 398ms (SD = 18), range = 376-419ms (see Figure 1—figure supplement 2). Overall, from this we can see that there was a high degree of overlap, at a group level, in the ranges of inter-beat intervals presented between the three conditions across the task. More importantly, at an individual level, mean ranges of interbeat-intervals calculated for each infant showed that the range of synchronous IBIs that any one infant experienced overlapped with the range of asynch-faster IBI’s by an average of 16ms, and with the range of asynch-slower IBIs by an average of 19ms.

Therefore, for our results to be an artefact of habituation, infants must have habituated to a changing range of heartbeat speeds presented during ‘synchronous’ trials (on average, 376-419ms) but not to two other fluctuating ranges of heartbeat speeds (361-392ms, and 399-436ms) with which the synchronous range overlaps. For this reason, we believe that it is very unlikely that habituation to an external stimulus characteristic such as absolute duration between beats could explain our results. However, we do consider it likely that our results are due to a novelty/habituation effect to a *relative* stimulus characteristic, namely the relationship between the stimulus and the infant’s heartbeat, and this is part of our central claim of the study, as our task was precisely designed to probe the infant’s sensitivity to interoceptive signals.

However, to further allay any concerns, we also conducted an additional analysis. We followed the suggestion of the reviewer and analysed only the first half of the synchronous trials. As predicted, the lack of power resulted in non-significance, but the looking times to synchronous trials were still numerically lower (4944ms, SD=3087) than those to slower (5925ms, SD=3250) and faster (5045ms, SD=3097) trials. However, as the reviewer suggested, looking duration did reduce over time for all conditions and therefore there was a confound between trial number, speed and synchrony.

[Editors' note: further revisions were requested prior to acceptance, as described below.]

*[…] The extra analysis you carried out to disentangle ERP from HEP is reported in the reply while this should appear in manuscript, as it is not about just convincing me or the reviewers but also the reader that might have the same issue in mind when reading your paper. This could be done briefly in the Results or Discussion sections and with further details in the supplementary material.*

We have now followed your advice and inserted our ERP analysis in the manuscript.